



# The first firn core from Peter 1st Island – capturing climate variability across the Bellingshausen Sea

Elizabeth R. Thomas[1], Dieter Tetzner[1], Bradley Markle[2,3], Joel Pedro[4,5], Guisella Gatacuia[6], Dorothea E. Moser[1,7], Sarah Jackson[8]

[1]Ice Dynamics and Paleoclimate, British Antarctic Survey, High Cross, Madingley Road, Cambridge, CB23 7XT, UK

[2]Department of Geological Sciences, University of Colorado, Boulder, USA

[3]Institute of Arctic and Alpine Science, University of Colorado, Boulder, USA

[4]Australian Antarctic Division, Kingston, Tasmania, 7050, Australia

[5]Australian Antarctic Program Partnership, Institute for Marine and Antarctic Studies, University of Tasmania, Hobart, Tasmania, 7004, Australia

[6]National Centre for Climate Research, Danish Meteorological Institute, Copenhagen, Denmark

[7]Department of Earth Sciences, University of Cambridge, Cambridge CB2 3EQ, UK

[8]Research School of Earth Sciences, Australian National University, Canberra, ACT 2600 Australia

*Correspondence to:* Elizabeth R. Thomas (lith@bas.ac.uk)

**Abstract.** Peter 1st Island is situated in the Bellingshausen Sea, a region that has experienced considerable climate change in recent decades. Warming sea surface temperatures and reduced sea ice cover have been accompanied by warming surface air temperature, increased snowfall, and accelerated mass loss over the adjacent ice sheet. Here we present data from the first firn core drilled on Peter 1st Island, spanning the period 2001-2017 CE. The stable water isotope data capture regional changes in surface air temperature, and precipitation (snow accumulation) at the site, which are highly correlated with the surrounding Amundsen-Bellingshausen Seas, and the adjacent Antarctic Peninsula (r>0.6, p<0.05). The unique in-situ data from an automatic weather station, together with the firn core data, confirms the high skill of the ERA5 reanalysis in capturing daily mean temperature and inter-annual precipitation variability, even over a small Sub-Antarctic Island. This study demonstrates the suitability of Peter 1st Island for future deep ice core drilling, with the potential to provide an invaluable archive to explore ice-ocean-atmosphere interactions over decadal to centennial timescales for this dynamic region.

## 1. Introduction:

The Sub-Antarctic Island of Peter 1st (Peter I Øy) is a former shield volcano (154 km$^2$), almost completely covered by a heavily crevassed ice cap. The islands' location in the Bellingshausen Sea, and just 450 km from the coast of West Antarctica, make it a scientifically important site for paleoclimate, ice sheet and oceanographic studies. The island is situated within the seasonal sea ice zone, in a region of the Southern Ocean that has experienced a rapid decline in sea ice cover in recent decades reaching a record low in February 2023 (NSIDC, 2023). The rate of sea ice decline in the Bellingshausen Sea since 1979 is comparable to the rate of ice loss in the Arctic (Parkinson, 2019). Reconstructions from ice cores suggest this recent change is part of a 20$^{th}$ century decline, evident in both proxy and observational based reconstructions (Abram et al., 2010; Porter et al., 2016; Thomas et al., 2019).

The closest landmass is the Antarctic Peninsula (AP) and Ellsworth Land coast, a region that has experienced considerable climate and glaciological change during the 20$^{th}$ century. Surface air temperatures on the AP, recorded at coastal research stations, have increased by as much as 2.5°C since the 1950s (Turner et al., 2005) constituting the largest warming in the Southern Hemisphere (Siegert et al., 2019). Despite a pause in the trend during the 21$^{st}$ century (Turner et al., 2016) warming has resumed to record levels (González-Herrero et al.,



2022) and paleoclimate archives suggest the warming during the late 20[th] century was part of a 100-year trend (Royles et al., 2013; Thomas et al., 2009; Thomas. and Tetzner., 2018), that is likely to continue in the future (Li et al., 2018). In addition to the rise in temperature, snowfall has increased dramatically during the 20[th] century (Thomas et al., 2015; Thomas et al., 2008; Thomas et al., 2017) attributed to changes in atmospheric circulation, sea ice changes and rising surface air temperature (Goodwin et al., 2016; Medley and Thomas, 2019; Porter et

al., 2016).

Glaciers along the Bellingshausen Sea and Ellsworth Land coast have retreated in recent decades; (Paolo et al., 2015; Pritchard et al., 2009; Smith et al., 2020). Many glaciers display dynamic thinning and grounding-line retreat that has been attributed to incursions of circumpolar deep water (CDW). The water in the Bellingshausen Sea is amongst the warmest in the Southern Ocean, with measured CDW temperature exceeding 1°C (Jenkins

and Jacobs, 2008). The island is situated to the northwest of the Belgica Fan, the culmination of the Belgica Trough, an exceptionally large paleo-ice stream. Ice sheet reconstructions suggest that during the last Glacial Maximum all the modern drainage basins along the Bellingshausen Sea coast were tributaries for a single large ice stream that may have extended to the continental shelf, less than ~200 km from Peter 1[st] Island. Thus, the location of Peter 1[st] Island, at the northern edge of the continental shelf, is of significance for both modern and

paleoclimate, oceanographic and ice sheet studies.

The first firn core from Peter 1[st] Island was drilled as part of the Sub-Antarctic Ice Core Expedition (SubICE), one of the projects of the international Antarctic Circumnavigation Expedition (ACE) 2017–2018 (Walton, 2018; Thomas et al., 2021). The aim of this study is to present the chemical and stable water isotope data from the Peter 1[st] Island ice core to determine its suitability for paleoclimate reconstructions. In addition, we utilise a

short instrumental record from an automatic weather station (AWS) from Peter 1[st] island to determine the skill of the ERA5 reanalysis data. We will 1) establish the firn core age-scale, 2) evaluate the skill of ERA5 at this island, 3) evaluate the firn core proxies against meteorological parameters from ERA5 and 3) discuss the suitability of this site for future deep ice core drilling.

**2. Data and methods:**

       **2.1. Ice core site**

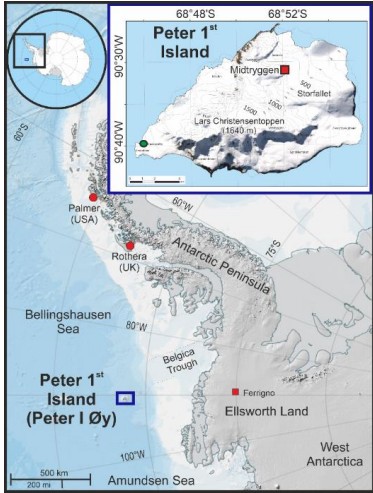

**Figure 1: Location of Peter 1[st] Island (Peter I Øy) in the Bellingshausen Sea, with the ice core site (blue rectangle) with closest Antarctic Peninsula research stations (red circles) and the Ellsworth Land ice core location referenced in**
**the text (Ferrigno, red box). Insert map showing the Peter 1[st] drilling location (red box), AWS (green dot) and island topography. Map produced using the Antarctic Digital Database, using data made available under the Creative Commons Attribution 4.0 International (CC BY 4.0) licence.**

In February 2017 a shallow ice core was drilled to a depth of 12.29 m on Peter 1[st] Island (68°51′05″ S, 90°30′35″ W). A site had been selected, based on satellite imagery, at the plateau at the top of the island (Lars



Christentoppen), however heavy cloud cover prevented helicopter landing at this site. Instead, a lower elevation site was found on a ridge (Midtryggen) at 730 m above sea level, in a small saddle on the eastern side of the island overlooking the main glacier Storfallet (Fig. 1). The snow surface was relatively smooth at this site (slope of ~5°), and ground penetrating radar (GPR) surveys were conducted in a ~500 m radius from the drill site (Thomas et al., 2021). Near continuous stratified layers were observed in the GPR profiles for the upper 14 m of

snow. The maximum time window for the GPR was set at ~43 m and bedrock was not detected at this depth, however, the full ice thickness has not been determined.

The firn core was drilled using a motorized Kovacs ice-core drill (Mark II) powered by a 4-stroke Honda generator, with core retrieval aided by a sidewinder winch. Ice core sections, with a maximum length of 80 cm, were stored in ethylene-vinyl-acetate-treated (EVA) polythene bags in insulated boxes.

The length and weight of each firn core fragment was measured to calculate density. Based on the measured density profile, and the Herron-Langway model, the estimated bubble close-off depth (when the firn air passages become closed at a density of 0.83 kg m$^{-3}$) is 34.5 m at this site. Visible melt layers >1 mm thick were recorded (Thomas et al., 2021), revealing an estimated 11% of the ice core is affected by melt, comparable to other Sub-Antarctic and coastal Antarctic ice core sites (Thomas et al., 2021), but considerably less than the Young Island

ice core (Moser et al., 2021). Discrete samples were cut at 5 cm resolution for ion-chromatographic (IC) and stable water isotope analysis, sealed in tritan copolyester jars.

### 2.2. Meteorological data

Meteorological data come from the European Centre for Medium-Range Weather Forecasts (ECMWF) ERA5 analysis (1979–2017) (Copernicus Climate Change Service, 2017), the fifth generation of ECMWF reanalysis.

ERA5 reanalysis currently extends back to 1950, providing hourly data at 0.25° resolution (~ 31 km). However, we note that the resolution of ERA5 may not fully capture local climate and precipitation on Peter 1$^{st}$ Island. An automatic weather station was located on the island between February 2006 and January 2007 (http://amrc.ssec.wisc.edu/aws/index.php?region=Ocean%20Islands&station=Peter%20I&year=2006, data downloaded 03/11/2020) providing a short (but incomplete) in-situ record of surface temperature (Thomas et al.,

2021). This data was not assimilated into the ECMWF model (de Rosnay, 2018). The AWS was located near 68°46.2S, 90°30.3W, at a height of ~128 m on the 'Radiosletta' Plateau on the NW side of the island (Fig.1, insert map). We utilise the observation-based Southern Hemisphere Annular Mode (SAM) Index of (Marshall, 2003) and the Southern Oscillation Index (SOI, defined as the normalized pressure difference between Tahiti and Darwin) of (Ropelewski and Jones, 1987).

### 110 2.3. Stable water isotopes

Isotopes, δ$^{18}$O and δD, were measured using a Picarro L2130-i analyser at the British Antarctic Survey (BAS), with an accuracy of 0.3 and 0.9 ‰ respectively. The measurements are reported against the international standard of Vienna Standard Mean Ocean Water (V-SMOW). Deuterium excess (d$_{xs}$) is the second-order parameter calculated from the two water isotope ratios (d$_{xs}$ = δD – 8*δ$^{18}$O) (Dansgaard, 1964).

### 115 2.4. Major ion chemistry

Major ion concentrations were measured using a high-performance Dionex Integrion ion chromatograph with an injection volume of 250 µL in a class-100 cleanroom at BAS. For the cation chromatograph, we applied a guard column type CS16-4µm (2 × 50 mm) and a CS16-4µm separator column (2 × 250 mm). For the anion chromatograph, we used an AG17-C guard column (2 × 50 mm) together with an AS17-C analytical column (2

× 250 mm). The chemical data presented here is for the purposes of annual layer counting. Ions include sulphate [SO$_4^{2-}$], methanesulphonic acid [MSA$^-$], Bromide [Br$^-$] and Sodium [Na$^+$], with an analytical precision, defined as the relative standard deviation of the lowest level standard, of 0.03, 0.07, 0.003, and 0.07 ppb respectively.

### 2.5. Snow accumulation

The annual snow accumulation is derived from the annual layer thickness (see section 3.1). The thickness is converted to meters of water equivalent (m w$_{eq}$ y$^{-1}$) based on the measured density. Thinning is corrected using the Nye model, which assumes thinning is proportional to vertical stress, appropriate for the upper 10% of the ice sheet. While the ice sheet thickness is unknown at this site, GPR confirms that bedrock is at least deeper than



43 m (Thomas et al., 2021). Given the sites elevation (730 m. a.s.l), and the relatively flat surface topography, a
depth of 130 m (ensuring the firn core bottom depth was within the top 10% ice thickness) is not unreasonable.
However, we acknowledge that this may not be the most appropriate thinning function for this site.

### 2.6. Estimating uncertainty

Uncertainty bars of 1 standard error (σ) are applied to all time series, except for 2006 (9.2 m) and 2013 (4.5 m)
where a 2σ value is applied to account for the influence of melt (section 3.2). For the predicted bottom age
estimate (section 4), the uncertainty estimate is calculated based on the difference between an upper and lower
snow accumulation estimate. The lower value is based on the snow accumulation derived using all 15 years and
the upper value derived with the two high melt years removed.

### 3. Results:

### 3.1. Age-scale

The age-scale has been derived using annual layer counting, based on the seasonal deposition of major ion
chemistry (Fig. 2). Given its maritime location, seasonal cycles are especially clear in chemical species with
marine origin, including $[SO_4^{2-}]$, $[MSA^-]$ and $[Br^-]$. These species relate to changes in marine productivity and
sea ice (e.g., Thomas et al., 2019 and references therein), which peak during the phytoplankton bloom in spring
and summer. Both $[SO_4^{2-}]$ and $[MSA^-]$ are robust seasonal markers in many coastal Antarctic ice cores
(Emanuelsson et al., 2022; Tetzner et al., 2022; Thomas and Abram, 2016), and have also proved to be valuable
for dating other sub-Antarctic ice cores (King et al., 2019; Moser et al., 2021).

Summer peaks were assigned if a consistent peak was observed in the marine ions (Fig. 2b-d). The stable water
isotope record (Fig. 2a) was used as a secondary tracer. An equal number of peaks are identified in the isotope
and ions records, however, there is often an offset in the location of the peak. For consistency, the location of
the major ion peak was used. Thus, the age is assumed to represent approximately December – November
corresponding to the summer sea ice break-up. The final age-scale extends from summer 2017 until summer
2002, encompassing 15 full years.

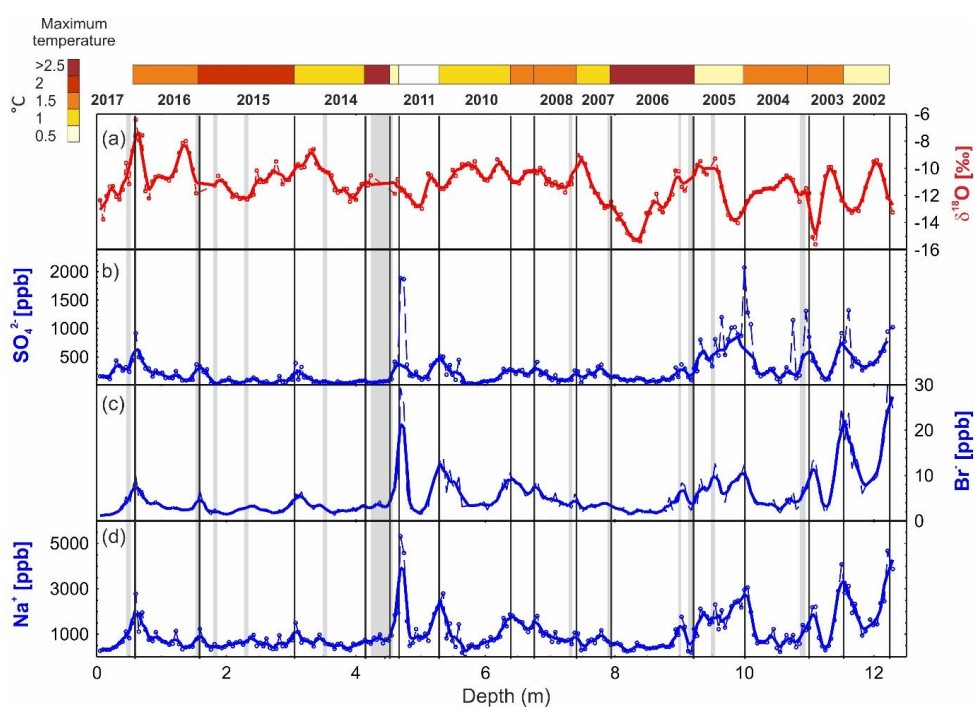



**Figure 2: Annual layer counting based on (a) Stable water isotopes (δ¹⁸O), (b) Sulphate (SO₄²⁻), (c) Bromide (Br⁻), and (d) Sodium (Na⁺) all plotted at 5 cm resolution (dashed curves) with a 3-point running mean (solid curves). Vertical black lines indicate the location of summer peaks. Vertical grey shading indicates observed melt layers of greater than 5 cm thickness. Top colour bar represents the maximum daily 2m temperature from ERA5.**

### 3.2. Evaluating the skill of ERA5 reanalysis to capture temperature

The Peter 1st firn core and the in-situ observations from an AWS provides a unique opportunity to evaluate the skill of the latest generation of reanalysis, ERA5, at a remote Sub-Antarctic Island. The hourly temperature data recorded by the AWS has been converted to daily averages for comparison (Fig. 3a). The AWS recorded intermittently from February to December 2006, and averages were only calculated on days when greater than 50% of the hourly data was available. There is exceptional agreement between the AWS and ERA5 daily data, with a correlation coefficient of r=0.91 for the days of overlap (n=149, p<0.0001). This demonstrates the high degree of skill in the ERA5 reanalysis data, although the period of comparison is short. The average daily temperature from the AWS data was -3.09 °C, 0.93 °C colder than the same days in ERA5.

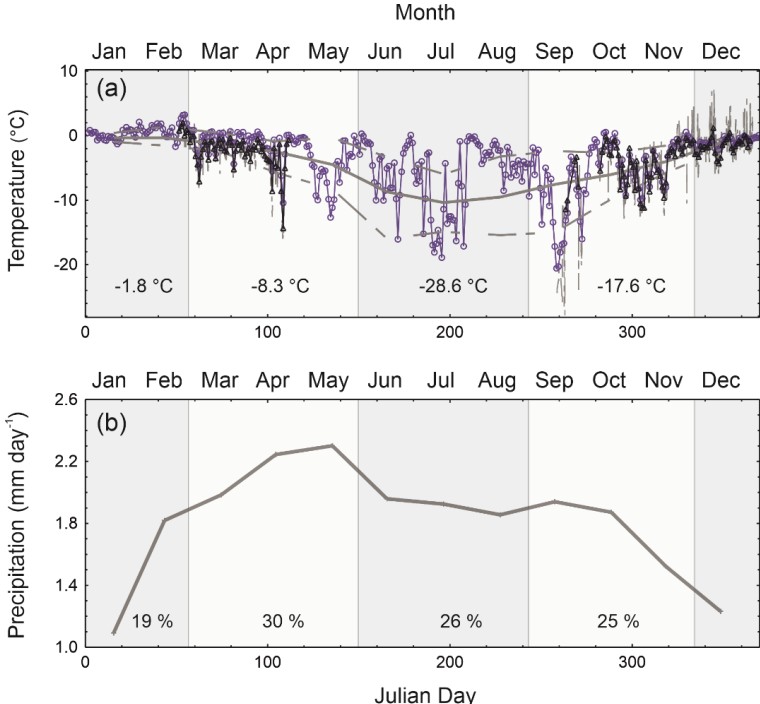

**Figure 3: Seasonal cycle in temperature (a) and precipitation (b) at Peter 1st island from ERA5 reanalysis (grey curve) (2002-2016 CE). Dashed grey curve represents 2.5 % and 97.5 % percentiles for temperature (a). Daily average temperatures during 2006 CE from ERA5 (blue circle) and AWS (black triangle), with hourly AWS temperature shown in thin grey curve. Grey shading highlights the seasons, with the corresponding average temperature and percentage of precipitation shown. AWS data available from University of Wisconsin-Madison Automatic Weather Station Program.**

The AWS was positioned at 128 m above sea level, while the ERA5 data is 2 m above sea level and the ice core was drilled at 730 m. To account for the adiabatic rate of temperature change for vertically moving air, a lapse rate is applied. The lapse rate varies dependent on the temperature and mixing ratio and is unknown for this site. A lapse rate of 0.68 °C/100 m was applied in Thomas et al., (2021) for the SubICE sites, based on a rate estimate for the western AP (Martin and Peel, 1978), assumed to be representative of the maritime locations.



However closer evaluation of the in-situ AWS data, and the interpretation of melt features at Peter 1[st], suggest this lapse rate may not be appropriate.

If the lapse rate of 0.68°C /100 m is applied the offset in average daily temperature between ERA5 (2 m) and the AWS (128 m) is just 0.07 °C. A recent evaluation of the bias between ERA5 and Antarctic temperatures suggested ERA5 has a warm bias of 0.58 °C for the AP and -0.66 °C for West Antarctica (Zhu et al., 2021). In addition, if the western AP lapse rate is applied to ERA5 this would suggest that daily mean site temperatures at the drill site never exceed -1.7 °C in the period 2002-2016. This does not fit with the evidence of visible melt features in the firn core.

If we assume that the multiple melt features at ~9.2 m depth correspond to the positive degree days during February 2006, then we would expect that site temperatures have exceeded 0 °C. Even if only for a few hours. The warmest daily average temperature from the AWS (February 2006) was 1.94°C. The appearance of melt features results from a dynamic interplay of atmospheric and snow conditions, but if melting generally occurs during positive temperatures, a minimum lapse rate of 0.32°C/100 m is required to allow for positive

temperatures at the drill site (602 m higher in elevation). This is lower than the moist adiabatic lapse rate (0.4-0.6 °C/100 m), and lower than lapse rates measured at the Sub-Antarctic Macquarie Island (Fitzgerald and Kirkpatrick, 2020). However, lapse rates as low at 0.31 °C/100 m have been reported at the Sub-Antarctic Marion Island (Nyakatya and McGeoch, 2008).

### 3.3. The relationship between temperature and surface melt.

An evaluation of the visible melt layers for this site suggests that 11 % of the total ice core is classified as melt (Thomas et al., 2021). This percentage is driven largely by a single 30 cm thick melt layer at a depth of 4.5 m (Fig. A1). Based on our annual layer counting, constrained by the $[SO_4^{2-}]$ peak at 4.6 m, this melt feature corresponds to the year 2013 CE. Over a period of 21 days in January 2013, the daily temperature (ERA5, not

corrected for elevation) remained above 0.5°C, reaching a maximum daily temperature of 2.5 °C (Fig. 2). The prolonged mild conditions likely explain the thick melt feature.

The second meltiest year on the record is 2006 CE. A maximum daily temperature of 3.2°C (ERA5) was recorded in February 2006, the warmest month in the record. Between February and March 2006, maximum daily temperatures exceeded 0.5°C for a total of 39 days. Positive temperatures during 2006 are corroborated by

AWS data, which recorded a maximum daily temperature of 1.94 °C. The highest hourly temperature recorded by the AWS was 7.1 °C, in December 2006 (Fig. 3).

A positive relationship is observed between maximum daily 2 m temperatures and melt thickness (r=0.4, p<0.1). Once the total melt thickness in each year is converted to a melt percentage (dividing the total melt thickness by annual layer thickness), the correlation with 2 m maximum temperatures increases to r=0.5 (p<0.1). The broad

alignment of the melt layers alongside the peaks in chemical species, provide additional evidence for a summer peak. The evidence that the warmest years on the record, 2006 and 2013 CE, correspond to the most melt affected sections in the core provide independent verification for the proposed age-scale.

### 3.4. Evaluating the skill of ERA5 reanalysis to capture precipitation.

In the absence of daily precipitation data from the AWS, we can use the snow accumulation derived from the firn core to evaluate the skill of ERA5 in capturing inter-annual precipitation variability. The average snow accumulation derived from the firn core (2002 – 2016) is 0.49 m weq yr[-1]. This is slightly lower that the estimated precipitation – evaporation (P-E) value of 0.55 m weq yr[-1] from ERA5 for this site (Fig. 4a). Case studies on the AP and coastal Ellsworth Land, using the previous generation of reanalysis products (ERA-

Interim and ERA-40), suggest that snow accumulation is underestimated by between 0.025 and 0.26 m weq per year (Thomas and Bracegirdle, 2009; Thomas and Bracegirdle, 2015). In this more maritime setting, and notwithstanding the different orographic positions of the firn core site on the island, there is an offset of approximately 6 cm per year (~12%) between the snow accumulation recorded in the firn core and ERA5 (P-E).

Snow accumulation is the sum of precipitation, evaporation, melt, erosion, and sublimation. Wind driven

erosion and re-distribution is estimated to remove between 5-20 cm yr [-1] of precipitated snow in Antarctic



coastal regions (Lenaerts and van den Broeke, 2012). Within the lower range of our observed offset. In addition, we have already established that this site is influenced by melt, which not only alters the density calculations, but may also suggest potential loss as melt run-off. Removing all years where the percentage of the annual layer thickness classified as melt exceeds 20% (years 2006 and 2013 CE), the revised snow accumulation (0.53 m weq yr$^{-1}$) is just 2 cm (~4%) lower than the ERA5 P-E.

### 3.5. Snow accumulation

Snow accumulation (2016 – 2002) at Peter 1$^{st}$ is positively correlated with P-E from ERA5. Strong correlations are observed over the island (r=0.75, p<0.01), with an extended zone of correlation (r>0.6) across the Bellingshausen Sea, the AP, and the Ronne-Filchner ice shelf (Fig. 4b). This relationship between snow accumulation at Peter 1$^{st}$ and the adjacent Ellsworth Land coast is confirmed by comparison with the snow accumulation record from the Ferrigno ice core drilled in 2010 (Thomas et al., 2015). Despite the short period of overlap (n=9), the two records are positively correlated (r=0.62, p<0.1) with a similar average snow accumulation rate (0.55 m w$_{eq}$ yr$^{-1}$ at Ferrigno).

A positive correlation is also observed between snow accumulation and surface air temperature from ERA5 (r=0.58, p<0.05). The spatial extent of the correlations (not shown) broadly mirrors the relationship with precipitation (Fig. 4b), extending from the Bellingshausen Sea over the AP.

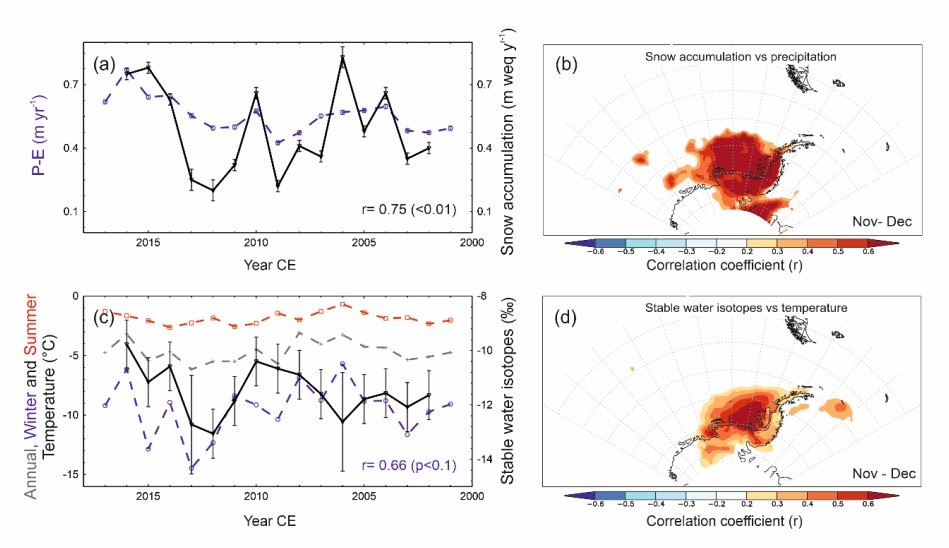

**Figure 4: (a) Peter 1$^{st}$ annual average snow accumulation (solid black), compared with precipitation-evaporation (P-E) (dashed blue) from ERA5 reanalysis data (2001-2017) with corresponding correlation coefficients (r) value shown. (b) Spatial correlation plot of annual snow accumulation with annual ERA5 P-E (all coloured areas p>0.05). (c) Annual average stable water isotope (δ$^{18}$O) (solid black), compared with 2 m temperatures from ERA5, as annual (grey dashed), summer (red dashed) and winter (blue dashed) averages. Correlation coefficient (r) between annual δ$^{18}$O and winter average 2m temperature after high melt years removed. (d) Spatial correlation plot of annual δ$^{18}$O with annual ERA5 2m temperature (all coloured areas p.0.05). All annual averages calculated as December to November, summer December to March and winter June-August. Uncertainty bars are one standard error (σ), for all years except 2006 and 2013, where 2σ are applied to account for additional uncertainties relating to melt.**

### 3.6. Stable water isotopes

The stable water isotope composition of Antarctic snowfall has been used to reconstruct past surface temperatures at annual to centennial timescales (e.g., (Stenni et al., 2017) and reference therein). However, the processes controlling isotopic composition are complex, relating to water vapour origin, distance from source



(Hatvani et al., 2017), condensation conditions, fractionation pathways (Markle and Steig, 2022), precipitation seasonality, intermittency and post-depositional changes (Münch et al., 2017; Fernandoy et al., 2012).

The annual average stable water isotopes (both δ[18]O and δD) at Peter 1[st] are weakly correlated with ERA5 2 m temperatures at the site (r=0.37, p>0.1) (Fig. 4c). Removing the two most melt affected years (2013 & 2006 CE) and comparing the annual record with winter average 2m temperature (June-August), increases the correlation with 2 m temperatures to r=0.61 (p<0.05) (Table 1) and r=0.66 (p<0.05) respectively.

The relationship between δ[18]O and temperature is much stronger over the adjacent ocean (Fig. 4d). The spatial correlation plot reveals a strong positive correlation with 2 m temperature over the Bellingshausen Sea (r>0.6),
within the approximate area of the seasonal sea ice zone.

### 3.7. Major ions

As expected for an island location, the major ions deposited at Peter 1[st] are largely of marine origin. The ratio of Cl[-]/Na[+] in the ice core is 1.8, consistent with the standard seawater ratio (1.79). Thus, at this site [Na[+]] can be considered primarily of marine origin (~95%). The Cl[-]/Mg[2+] and Cl[-]/Ca[2+] ratios suggest that seawater accounts
for 82% and 73% of the [Mg[2+]] and [Ca[2+]] concentration respectively.

The average [Na[+]] at Peter 1[st] is 998 ppb, consistent with a coastal location. A database of 105 Antarctic ice cores (Thomas et al., 2022) suggest that the highest [Na[+]] in an Antarctic ice core is observed on the Fimbul ice shelf, coastal East Antarctica, where average concentrations exceed 2700 ppb. The average [Na[+]] at Peter 1[st] is higher than the Sub-Antarctic Island of Bouvet, in the South Atlantic, where the average [Na[+]] was 101 ppb
(King et al., 2019). It is also higher than values on the AP, which range between 50-215 ppb (Emanuelsson et al., 2022; Thomas et al., 2022), however, the drill sites are higher in elevation and further from the oceanic source.

The average [SO$_4^{2-}$] at Peter 1[st] is 267 ppb. This is considerably higher than concentrations found in AP ice cores, where values of between ~30 and 70 ppb are observed (Thomas et al., 2022; Emanuelsson et al., 2022).
However, higher concentrations are observed at Bouvet Island (King et al., 2019) (529 ppb) and the Fimbul ice shelf (536 ppb) (Thomas et al., 2022).

### 3.8. Source regions and transport pathways of snowfall

To establish the potential source regions of proxies, and their transport routes to the site, we evaluate the snow accumulation against climate parameters from ERA5 and common climate indices (Table 1). However, we
acknowledge that the short duration of the ice core record may lead to spurious or not statistically significant results. Thus, we have generated a "pseudo core" based on the P-E extracted from ERA5 at the firn core location. Spatial correlations are run using the 15 years of firn core data, the same 15-year window (2002 – 2016) for the pseudo core and the extended 37 years (1979 – 2016) of pseudo core data (Fig. 5).

The spatial correlation between snow accumulation (P-E for the pseudo cores) and geopotential height (850 hPa)
are shown in figures 5a, 5d and 5g. A distinct pattern of positive correlations is observed over the southern tip of South America and the Drake Passage, with opposing negative correlations over the Atlantic and Indian sectors of the Southern Ocean. The spatial correlations are remarkably similar when using both the firn core derived snow accumulation and the pseudo core data over the same period (Fig. 5d). Over the extended period (1979–2016) the positive correlation remains over southern South America, suggesting this feature is robust over the
37-year period (Fig. 5g). However, the region of negative correlations shifts from over the ocean to over the Antarctic continent, in a pattern reminiscent of the Southern Annular Mode (SAM) (Marshall, 2003).

Meridional winds draw warm-moist air from the South Pacific sector of the Southern Ocean as shown by the strong correlations between snow accumulation and meridional winds (blue area in Fig. 5b, 5e and 5h). The blocking high- and low-pressure anomalies to north and south of the AP funnel the air-masses from the
Bellingshausen Sea, over the AP and into the Weddell Sea sector. This zonal transport is shown by the correlations between snow accumulation and westerly winds (red area in Fig. 5c, 5f, and 5i) across the AP. Thus, enhanced precipitation (more snow accumulation) is associated with stronger northerly and westerly winds across the Amundsen and Bellingshausen Seas. This spatial pattern in meridional winds is observed for the firn core period (2002-2016) and maintained over the longer period (1979-2016), suggesting a degree of
stability over decadal timescales. However, there is a definite southward shift in the correlations with zonal





winds. During the shorter period of the firn core record, the strongest correlations are with winds at the same latitude or slightly south of the island. However, during the extended period (1979-2016) (Fig. 5i) the highest correlations are observed to the north of the island, further away from the coast of West Antarctica.

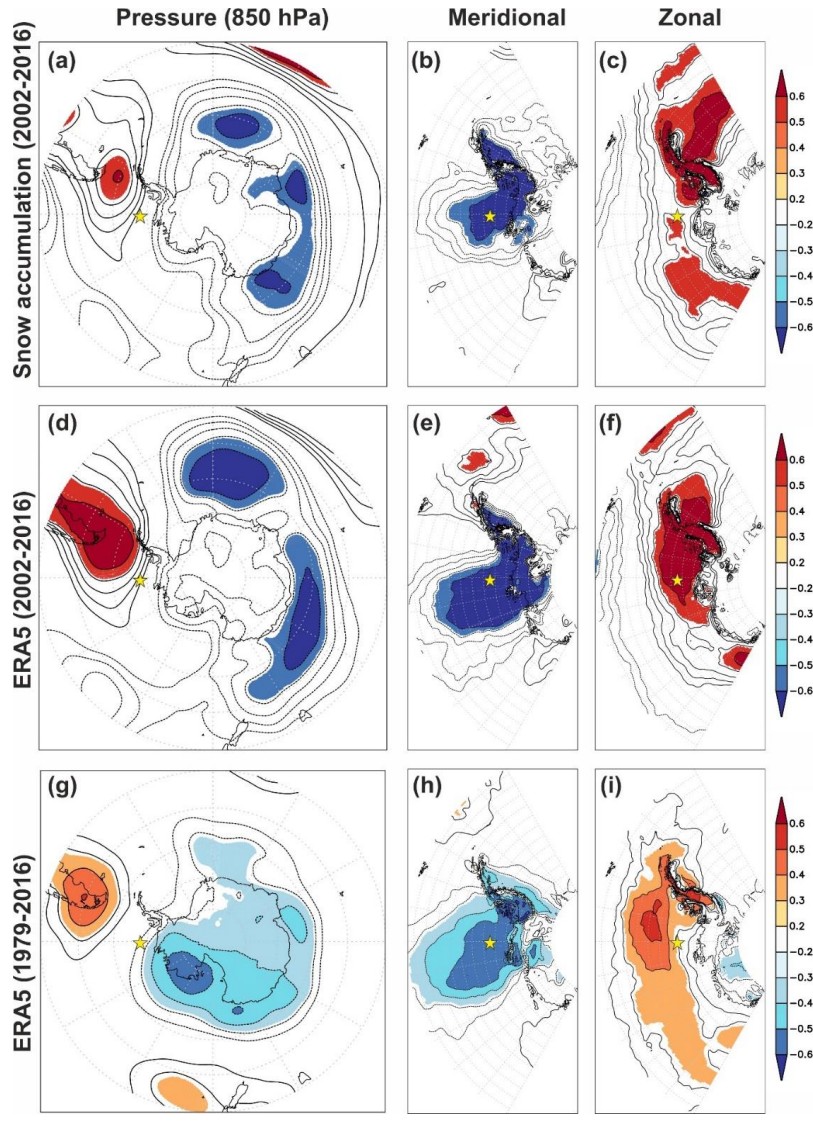


**Figure 5. Exploring the mechanisms driving precipitation variability at Peter 1st Island (yellow star). Spatial correlation between Peter 1st snow accumulation and (a) geopotential height (850 hPa), (b) meridional winds and (c) zonal winds (ERA5, 2002-2016). Plots d-f are the same spatial correlations but using a timeseries of P-E extracted from ERA5 for the ice core site ("pseudo cores"). Plots g-i are the**
**same but for the extended period 1979 – 2016.**

The snow accumulation is positively correlated with the SAM (r=0.59, p<0.05), with a comparable correlation to that when using the pseudo core data (ERA5 P-E, r=0.55. p<0.05). The correlation with SAM becomes weaker when using the extended pseudo record (1979 – 2016), r=0.3 (p<0.1). Snow accumulation is negatively



correlated with the Southern Oscillation Index (SOI), r=-0.48 (p<0.1). The SOI is the normalized pressure
difference between Tahiti and Darwin, which tracks the atmospheric component of El Niño Southern Oscillation
(ENSO). This suggests that snowfall on the island (and potentially the proxies contained therein) is influenced
by large-scale modes of atmospheric variability and tropical teleconnections. However, the correlation between
the pseudo core (P-E) and SOI is not significant (p>0.1) over the 15-year period of the firn core (2002-2016), or
the extended period (1979-2016). This suggests that the relationship with both climate indices is not temporality
stable as observed in other studies (e.g., Thomas et al., 2015). We do not find any correlation between $\delta^{18}O$ with
either SOI or SAM. The correlation is not improved by removing the highest melt years (2013 & 2006). There is
also no correlation between 2 m temperature (ERA5) with either SOI or SAM, suggesting that temperatures at
the site are not influenced by large-scale modes of variability, at least over short timescales.

**Table 1: Comparing the annual average 2 m temperature (2 m Temp) and P-E data from ERA5 with the
annual average stable water isotopes ($\delta^{18}O$) and snow accumulation data from the Peter 1$^{st}$ firn core. The
2 m Temp range at the firn core location (730 m a.s.l) is calculated based on a lower lapse rate of 0.31 °C/
100 m and an upper lapse rate of 0.68 °C/ 100 m (Thomas et al., 2021). Correlation coefficient between
SAM, SOI, P-E and 2m temperature are shown only if p < 0.05 (n=15). The correlations when the two
melt years (2006 & 2013) are removed are shown in brackets (n=13).**

| Species | Mean | Max | Min | SAM (r) | SOI (r) | ERA5 (P-E) (r) | ERA 2m Temp (r) |
|---|---|---|---|---|---|---|---|
| **ERA5** | | | | | | | |
| 2m Temp (°C) (Sea level) | -4.6 | -3.0 | -6.2 | | | 0.60 | |
| 2 m Temp (°C) (730 m a.s.l) | -9.4 to -6.8 | -7.8 to -5.3 | -11.0 to -8.4 | | | 0.60 | |
| P-E (m weq) | 0.55 | 0.77 | 0.43 | 0.55 | | | 0.60 |
| **Peter 1$^{st}$ firn Core** | | | | | | | |
| $\delta^{18}O$ (‰) | −11.3‰ | −6.4 ‰ | −15.6 ‰ | | | | (0.61) |
| Snow accumulation m weq | 0.49 (0.53) | 0.83 (0.77) | 0.20 (0.22) | 0.59 | -0.48 | 0.75 (0.85) | 0.58 |

## 4. Discussion:

The objective of this study is to establish if a firn core from Peter 1$^{st}$ Island is suitable for paleoclimate
reconstructions. Here we discuss the climatological data captured by the firn core and explore this sites potential
for future deep ice core drilling.

### 4.1. Annual layer counted age-scale.

We have established that seasonal cycles in major ion chemistry and $\delta^{18}O$ are suitable for annual layer counting.
A prominent peak in major ion chemistry (including [$SO_4^{2-}$]) at 4.6 m depth corresponds to the Puyehue-Cordon
Caulle eruption from southern Chile. This VEI5 rated eruption began in June 2011 and appears in our record
from late 2011 and into 2012 and provides at least one independent reference horizon. High biogenic [$SO_4^{2-}$]
background can make identification of volcanic [$SO_4^{2-}$] peaks difficult, as demonstrated at Antarctic Peninsula
sites (Emanuelsson et al., 2022; Tetzner et al., 2021). While the average [$SO_4^{2-}$] is lower at Peter 1$^{st}$ than many
coastal Antarctic sites, the background [$SO_4^{2-}$] may make identification of bi-polar eruptions difficult, and the
presence of volcanic ash (tephra) may be required to identify volcanic tie points in a deeper ice core from this
site. The lower end of the age-scale is constrained by the absence of the recently identified Sturge island
eruption in 2001 (Tetzner et al., 2021). This Sub-Antarctic eruption has been detected as large shards in other
Sub-Antarctic islands (Moser et al., 2021) and the Ellsworth Land coast adjacent to Peter 1$^{st}$. Although not
definitive, if our age-scale extended beyond 2002, we might expect to see some evidence of this eruption at this
site.



The bottom age of 2002 CE is within the uncertainty range of a previous estimate (2004 +/- 2 years), based on the fitted density profile using the P-E from ERA5 (Thomas et al., 2021). When using the newly calculated snow accumulation (0.49 m weq yr-1), the updated density derived bottom age is 2002 +/- 1 years. Thus, validating the use of the densification model to estimate the potential bottom age for a deeper ice core. The density profile estimate suggests that an ice core drilled to the maximum GPR layer depth (identified at 43 m

(Thomas et al., 2021)), would provide a record dating back to 1951.2 CE +/- 5 years. While the ice sheet thickness remains unknown, based on the measured snow accumulation rate and density profile, we conclude that an ice core drilled to 100 m depth would provide a record dating back to 1833 CE +/- 13 years.

### 4.2. Proxy validation and comparison with reanalysis (ERA5)

With only a single AWS temperature record, that does not comprise a full year and is some distance from the ice

core site, we must rely on the reanalysis data to evaluate our proxy measurements. The previous ECMWF reanalyses products (ERA-Interim and ERA-40) have been tested widely in Antarctica (e.g., (Bromwich and Fogt, 2004)) and at sites in the adjacent Ellsworth Land coast and the AP (Thomas and Bracegirdle, 2009: 2015). A recent study confirms that ERA5 accurately captures variability across Antarctica (Zhu et al., 2021) and in near-surface air temperature and wind regimes over the adjacent AP (Tetzner et al., 2019).

The high correlation between daily mean 2 m temperature in ERA5 and temperature recorded in an AWS demonstrate the high degree of skill in ERA5 at this location. The correlation of 0.91 is consistent with the correlation of 0.92 determined when comparing observations and ERA5 annual mean temperatures for Antarctica (Zhu et al., 2021). However, in the absence of a measured lapse rate it is difficult to determine the true temperature bias in the ERA5 reanalysis for Peter 1[st]. If we apply the lapse rate of 0.32 °C/100 m (proposed

in section 3.2) this suggests that ERA5 has a cold bias of 0.37°C.

Comparing the 15 years of annual mean snow accumulation from the firn core with P-E from ERA5 (Fig. 4a) revealed a high temporal correlation between the two records (r = 0.75, p<0.01; Table 1) and comparable absolute values. The slight over-estimation of the ERA5 total P-E (~ 4-6 %) is less than the offset observed at adjacent sites on the AP (Thomas and Bracegirdle, 2009:2015). The slight offset is likely an artefact of the

resolution of the ERA5 reanalysis data (0.25° resolution (~ 31 km)), which is not sufficient to differentiate Peter 1[st] island from the surrounding ocean. However, despite this limitation our study suggests that ERA5 displays a high degree of skill in capturing absolute amount and temporal variability in precipitation changes, at this firn core site on a small and mountainous island location. Importantly, the spatial correlation maps reveal that the high correlation between snow accumulation and ERA5 P-E extends over the AP and Amundsen-

Bellingshausen Sea region (Fig 4b). These results support the use of the firn core for regional climate reconstructions.

### 4.3. Relationship between snow accumulation and stable isotopes with precipitation and temperature

The strong correlation (r=0.75, p<0.01; Table 1) between annual snow accumulation and P-E in the corresponding ERA5 grid cell over the common 15-year interval suggests that the ice core layer thickness (snow

accumulation) is dominated by changes in precipitation. Thus, a longer reconstruction could provide valuable insight into changes in snow accumulation and surface mass balance across a large and dynamic region of Antarctica. Although traditionally viewed as a proxy for past surface temperature, the annual average $\delta^{18}O$ at this site is not correlated with site annual average ERA5 2 m temperature (0.37, p=0.17). However, the observed weak correlation coefficients ($\delta^{18}O$ vs 2m surface air temperature) are consistent with ice cores from the AP and

coastal Ellsworth Land (e.g., Thomas et al., 2009; 2013). The correlation between $\delta^{18}O$ and temperature is improved by removing the melt years and correlated with winter average 2m temperatures. This may suggest that winter conditions play a more dominant role in modulating $\delta^{18}O$ at this site, or that the summer $\delta^{18}O$ signal is weaker or been lost. The summer months (December – February) receive the lowest amount of snowfall (Fig 3b), just 19% of the total. Thus, the isotopic signal (which is precipitation biased) will be more strongly

weighted to the spring, winter, and autumn months respectively. In addition, the summer months may experience more melting, potentially smoothing the isotopic signal of the summer snow deposits.

At Peter 1[st], the annual snow accumulation is strongly related to annual ERA5 2m surface air temperatures. The positive correlation between snow accumulation and ERA5 2m T (r=0.61, p<0.01) reflects the relationship between temperature and the saturation water vapour pressure governed by the Clausius–Clapeyron relation.

This relationship has been observed at ice core sites across the AP (e.g., Thomas et al., 2017), confirmed at the



continental scale (Medley and Thomas., 2019) and in a data-assimilation approach using global circulation models (Dalaiden et al., 2021). In the correlation map (Fig. 4d), there is a significant region of correlation between $\delta^{18}O$ and 2 m temperatures over adjacent ocean, especially within the seasonal sea ice zone. This suggests that sea ice may play a role in modulating $\delta^{18}O$ at this site. Sea ice has been shown to directly alter
$\delta^{18}O$, through an enrichment of the water vapour (Bromwich and Weaver, 1983). A reduction in the length of the sea ice season and an overall decline in sea ice coverage in the Amundsen and Bellingshausen Seas has been attributed to the warming trends observed in previous West Antarctic reconstructions (Küttel et al., 2012; Steig et al., 2009; Thomas et al., 2013). Thus, we conclude that over longer timescales $\delta^{18}O$ and snow accumulation from any future ice cores from this site will capture changes in surface air temperatures in this region.

**4.4.   Drivers of variability and the influence of atmospheric modes**

The record is too short to draw robust conclusions about the role of large-scale atmospheric circulation. However, given the islands location we expect this site to be strongly influenced by the Amundsen Sea Low (ASL), a climatological low-pressure system that exerts considerable influence on the climate of West Antarctica (Hosking et al., 2013). Enhanced northerly flow over the Bellingshausen Sea during the positive
phase of the SAM has been attributed to the large increase in snowfall during the late 20[th] century (e.g., Thomas et al., 2017; Medley and Thomas, 2019). The observed relationship between annual snow accumulation at Peter 1[st] and annual meridional winds over the Bellingshausen Sea replicates the pattern seen at several AP sites (Thomas et al., 2008; 2015; 2017). This shared transport route is confirmed in the back trajectory analysis from the AP sites, which are dominated by air masses that cross directly over Peter 1[st] Island (Thomas and
Bracegirdle, 2015).

Pressure in the ASL region is strongly modulated by large-scale modes of variability, especially SAM and ENSO (Fogt et al., 2012; Hosking et al., 2013). Despite the short period investigated, the snow accumulation does display significant relationships with both SAM (positive) and ENSO (negative). However, we are unable to ascertain the stability of this relationship, which has been proven to vary temporally at many sites across West
Antarctica (Thomas et al., 2015; Wang et al., 2017). Indeed, the spatial correlation of annual average P-E from the Peter 1[st] site (ERA5 "pseudo core") displays a stronger and more distinct spatial SAM pattern when using the longer record (Fig. 5g) than the period from 2002-2016 captured by the firn core. However, the correlation between annual SAM and annual average P-E in the 37-year pseudo core is lower than the 15-year period captured by the firn core, r=0.30 and r=0.55 (Table 1) respectively. This likely reflects the non-stationarity of
the correlations between SAM and snow accumulation. Since the late 1990s, and the period captured by the firn core, the SAM has been predominantly in its positive phase, while the period from 1979-2016 is characterised by a shift from negative to positive SAM phase (Marshall, 2003).

While we might expect the ASL to influence the snow accumulation at Peter 1[st], the relationship with snow accumulation and geopotential height in this region is not particularly strong. Instead, the Peter 1[st] annual snow
accumulation (and the pseudo core data) appears more strongly influenced by the high-pressure anomalies over the Drake Passage, to the north of the AP. Long-lived, and relatively stationary anticyclones have been shown to influence snow accumulation over West Antarctica (Emanuelsson et al., 2018) by impeding the westerly circulation. These anticyclones, also known as blocking events, can deflect marine air towards to continent, resulting in increased precipitation.  This may also reflect a shift in circulation patterns, as the observed trend
toward deeper sea level pressures over the ASL region during the late 20[th] century has become less pronounced during the early 21[st] century. Indeed, this shift in circulation patterns has, in part, been attributed to the slowdown in the warming trend observed in the AP (Turner et al., 2016).  Therefore, over longer time periods we might expect that snow accumulation from a future deep ice core would capture changes in ASL and blocking event variability.

**4.5.   Drivers of surface melt and the impact on proxy preservation**

Despite its maritime location, Peter 1[st] island is situated south of the polar front at a comparable latitude to much of the East Antarctic coastline (~70°S). While the annual average temperature (1979-2017) is -9.5 °C, with summer temperatures of -5.1 °C (Thomas et al., 2021), the daily temperatures from ERA5 indicate maximum temperature at the site has exceeded 3°C. This maximum in February 2006 was verified by in-situ observations
from an AWS. Over the 15-year period (~5500 days) covered by the firn core, there were a total of 189 positive degree days. Many, but not all, of these positive degree days correspond to visible melt layers in the ice core. However, there are some notable exceptions where melt features do not coincide with positive degree days.



Many of the major melt periods also coincide with documented evidence of atmospheric rivers (ARs). These narrow bands of enhanced water vapour transport heat and moisture from the mid- to the high-latitudes and have
been attributed to melt events across West Antarctica (Nicolas et al., 2017). Wille et al (2019) derived an AR detection algorithm to demonstrate that between 40-80% of surface melt on the western AP (1979-2017) is attributed to ARs that make landfall during the winter months (March-October). Many of the ARs identified in that study pass directly over Peter 1st Island and may explain the occurrence of visible melt features in the Peter 1st firn core during the winter months. The yearly percentage of AR occurrences calculated in Wille et al (2019)
reveal that two of the most abundant AR years, 2006 and 2013, correspond with the strongest melt features in the firn core. The year 2010 also contained a high number of AR occurrences, however, this year does not correspond to any major melt features in our record. However, 2010 was a high snow accumulation year and the enhanced moisture transport characterised by ARs is also known to increase precipitation. The physical mechanisms relating ARs to surface melt are complex, and thus it may be possible that some ARs passing over
Peter 1st result in increased precipitation, but not visible melt features. Or that the ARs during 2010 that made landfall in West Antarctica did not pass directly over Peter 1st island.

It is only during the most extreme years that melt has had a notable impact on the proxy preservation. A high-resolution evaluation of the melt features, and their impact on chemical elution, is subject to further study. However, there is some evidence that the chemical and isotopic records during the extreme melt event in 2013
have been altered. This is observed in the near homogeneous concentrations of major ion and $\delta^{18}O$ values during this melt feature, which appear to have removed the seasonal signal. The occurrence of melt has also likely increased the uncertainty in the snow accumulation calculations. For example, stronger correlations between snow accumulation (and $\delta^{18}O$) and precipitation (and temperature) are achieved when the two highest melt years (2013 and 2006) are excluded.

Instrumental records from the AP suggest that annual surface air temperatures have increased by approximately 2.5°C since the 1950s (Turner et al., 2005), with reports of record-breaking heatwaves in recent years (González-Herrero et al., 2022). This is corroborated by ice core records across the AP and coastal Ellsworth Land, which suggest a prominent warming trend during the latter half of the 20th century (Thomas et al., 2013; Thomas et al., 2009; Thomas. and Tetzner., 2018). Despite the absence of a significant warming trend during the
21st century (Turner et al., 2016), the temperatures during the 21st century are still considerably warmer than the early and mid-20th century. Thus, we might expect that the melt frequency observed during this period (2002-2016) will also be much higher than at any time in the recent past. A deeper ice core drilled from this location may be subject to melting in the surface layers, due to continued regional warming (González-Herrero et al., 2022), although the impact of melting is likely limited to the mid-20th century onwards.

This hypothesis is supported by the melt history obtained from the James Ross Island (JRI) ice core, in the north-eastern tip of the AP (Abram et al., 2013). Mean annual temperature of −14.31°C were reported at the JRI site during the 1980s (Aristarain et al., 1987) however, during the period 2001-2017 the annual average temperature increased to -7.5°C, warmer than the -9.5°C observed at Peter 1st. This may explain why the average melt layer thickness of 3.2 cm per year at JRI is higher than the observed 1.8 cm per year at Peter 1st.
The visible melt features at JRI display a clear acceleration in frequency during the late 20th century (Abram et al., 2013). However, this melt has not had a notable influence on the proxy preservation or subsequent paleoclimate reconstructions generated from this site (e.g., Abram et al., 2013).

### 5. Conclusions:

Here we present the first climatic interpretation of $\delta^{18}O$, and snow accumulation data contained in a firn core
drilled on the remote sub-Antarctic Island of Peter 1st. We conclude that a deep ice core from this site has the potential to provide valuable paleoclimate reconstructions, exploring the ice-atmosphere-ocean interactions in the Bellingshausen Sea based on the following findings:

- The firn core can be annually layer counted and verified using a volcanic reference horizon from the Cordón Caulle eruption in 2011.
- The ERA5 reanalysis displays a high degree of skill at reproducing site surface temperature and snow accumulation. This is confirmed by comparing daily temperatures from an AWS against ERA5 2-m temperatures and by comparing annual average snow accumulation from the ice core against annual average precipitation (P-E) from ERA5 at the corresponding grid cell. Thus, demonstrating that ERA5 can capture

climate variability even at a small sub-Antarctic Island and supporting the use of ERA5 as a suitable dataset to interpret climate proxies in this firn core.

- Snow accumulation observed in the firn core is significantly correlated to both regional precipitation (P-E) changes and changes in surface air temperature.
- Snow accumulation at the firn core site is likely related to large-scale modes of atmospheric variability, including SAM. However, the stability of the relationship between SAM and snow accumulation cannot be
confirmed beyond the 15-year interval of the firn core.
- The $\delta^{18}O$ record, although weakly correlated with site temperature, displays a strong and significant relationship with air temperature over the seasonal sea ice zone in the Bellingshausen Sea.
- The melt frequency is lower than observed at existing deep ice core sites from coastal Antarctica. The melt features broadly correspond to temperature, with the two most extreme melt years (2006 and 2013)
coincident with high temperatures and the documented occurrence of atmospheric rivers.
- The GPR data from this site reveal near continuous stratified layers in the upper 43 m (Thomas et al., 2021).
- While the ice sheet thickness remains unknown, based on the measures snow accumulation rate and density profile we conclude that an ice core drilled to 100 m depth would capture climate variability of the past ~200-years.


**Appendices:**

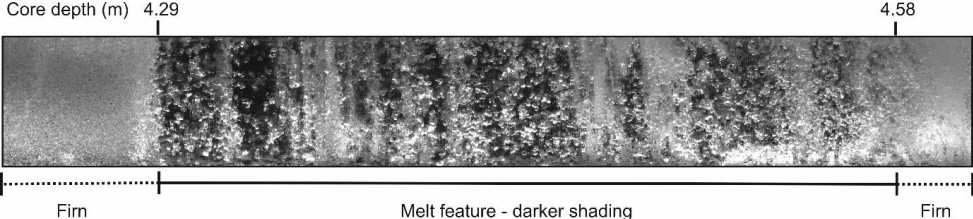

Core depth (m)   4.29                                                                 4.58

Firn                         Melt feature - darker shading                         Firn

**Figure A1: Line scanned image of the Peter 1ˢᵗ firn core. Highlighting the prominent melt feature,**
**observed as a dark area (higher density) between 4.29-4.58 m depth, compared to the lighter shaded (lower density) areas corresponding to firn.**

**Data availability:**

All data is submitted to the UK polar data centre (DOI pending) or available on request (lith@bas.ac.uk).


**Author contributions:**

ET designed the project and prepared the manuscript with contributions from all authors. JP, BM, GG conducted the fieldwork, DT produced the age-scale, DM contributed to the sample preparation, SJ conducted the IC analysis.

**Competing interest:**

ET is an editor of Climate of the Past. The peer-review process was guided by an independent editor, and the authors have also no other competing interests to declare."

**Acknowledgements:**

Funding was provided to subICE by École Polytechnique Fédérale de Lausanne, the Swiss Polar Institute, and
Ferring Pharmaceuticals Inc. ERT received core funding from the Natural Environment Research Council to the British Antarctic Survey's Ice Dynamics and Palaeoclimate programme. Joel B. Pedro acknowledges support



from the European Research Council under the European Community's Seventh Framework Programme (FP7/2007e2013) and ERC grant agreement 610055 as part of the ice2ice project and from the Australian Government Department of Industry Science Energy and Resources, grant ASCI000002. We are grateful to the
Norwegian Polar Institute for granting us permission to visit Peter 1st island. The authors appreciate the support of the University of Wisconsin-Madison Automatic Weather Station Program for the data set, data display, and information, NSF grant number 1924730 and ECMWF for providing ERA5 reanalysis data. We thank Laura Gerish (BAS) for producing the maps. We thank Joe Brown and Daniel Emanuelsson for the line scanning image presented in A1. Data used in this study are available through the UK Polar Data Centre. The authors
would like to acknowledge the coordinators and participants of the Antarctic Circumnavigation Expedition for facilitating collection of the subICE cores, especially David Walton, Christian de Marliave, Julia Schmale, Robert Brett, Sergio Rodrigues, Francois Bernard, Amy King, Roger Stilwell, and Frederick Paulsen.

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
