# Peer review of "The first firn core from Peter 1st Island – capturing climate variability across the Bellingshausen Sea"

_EGUsphere, 2023_

## Author Comment (AC1)

Response to Reviewer 1

The reviewer comments in black and our response in blue.

Reviewer 1

Review of "The first firn core from Peter 1st Island - capturing climate variability across the Bellingshausen Sea" by Thomas, et al. for Climate of the Past

Thomas et al. provide the initial results from a firn core retrieved from Peter 1st Island. The stable water isotopes and snow accumulation are compared with reanalysis data which they also ground-truth with meteorological data from an automated weather station. The authors provide evidence of some compelling connections with regional/large-scale climate variability to support further drilling campaigns in the area. My primary concerns rest with the low sample number used in the correlation analyses and the influence of substantial melt in this shallow firn core. Although the authors address these limitations to a degree, I would offer a few suggestions or clarifications that will hopefully help bolster their results and interpretation. Nevertheless, the authors demonstrate the potential that a longer core from this site could provide to understand past climate variability in the Bellingshausen Sea region.

Thank you for taking the time to review this manuscript. We agree with the limitations of short datasets and appreciate your suggestions for improvements.

Major revisions

Many of the conclusions from this manuscript are reliant on correlation analysis and standard p-values. However, only 15 years of data are available, and only 13 when two melt outliers are removed. Although I am inclined to trust the relationships discussed in the paper given the strong correlation values, I do wonder about "significance". The authors attempt to reconcile this issue by creating a pseudo-core data. I worry about the pseudo-core based solely on the ERA5 data which is then correlated with additional variables from ERA5. Could the interpolation schemes used in ERA5 generate spuriously high correlation coefficients? For instance, are there inherent correlations among adjacent or proximal grid cells that may be coming to light here. Circumventing the inability to collect more data at this stage (although I very much hope you are able to obtain a longer core from this site in the future), perhaps applying a bootstrapping method to the correlation analysis would provide somewhat more robust p-values.

We agree that the short duration of these cores is a limitation. The intention was not to over-state the correlations for a short record, but to demonstrate the potential of this site for future deep drilling. We will add additional clarification to the text to acknowledge the limitations, including the potential to generate higher correlation coefficients when comparing ERA5 variables with each other.

Thank you for the suggestion to use bootstrapping. We have applied this to our data, using 2000 samples, to provide more robust confidence intervals. This will be updated in table 1.

The melt in the core is also a concern which relates back to the robustness of the correlation analysis. Given such a short record, the disruption to the isotope, accumulation, and chemistry records by the melt events could greatly alter the results. The authors do address this by removing the two years of greatest melt in some analyses. How much does the melt influence preceding years? For instance, the year 2012 is almost non-existent. Could its inclusion in the correlation analysis skew results? However, once you start eliminating every year potentially altered by melt, you would not

have many years left in your analysis. The authors make a good point that melt events are likely more pronounced in recent years and would likely be fewer in frequency in a deeper core. This prediction supports the need for a deeper core but does not address the issues in the interpretation of the results for this manuscript. The data are what the data are; there's no changing that. Is there a way to ensure the results are more robust? In the case of melt affecting the time scale, could making a stronger case for the volcanic marker help here (see minor revisions)? Serendipitously, the second melt event in 2006 is captured by the AWS data. Could that be utilized to provide support for the time scale and subsequent analyses? I apologize that I don't have great suggestions for how to overcome these limitations.

Thank you again for suggesting improvements and acknowledging that we cannot change the data we have. We have tried to incorporate independent tie-points to strengthen the age-scale, however, with such a limited time span we don't have many (e.g., eruptions). We have restructured the section regarding volcanic tie points, so that it follows the results for the age-scale (3.1). Presenting the data together will hopefully strengthen the results.

You correctly point out that the melt is likely influencing subsequent years (e.g., 2012). However, as noted, this record is limited by the lack of data points and removing more would further reduce the statistical significance. The importance and significance of melt is discussed in a recent review [Moser et al., 2023], and we will reference this study in the text to improve our discussion regarding melt.

Minor revisions

Lines 56-60 - citation needed

Ó Cofaigh, C., Larter, R. D., Dowdeswell, J. A., Hillenbrand, C.-D., Pudsey, C. J., Evans, J., and Morris, P. (2005), Flow of the West Antarctic Ice Sheet on the continental margin of the Bellingshausen Sea at the Last Glacial Maximum, J. Geophys. Res., 110, B11103, doi:10.1029/2005JB003619.

Line 68, should be 4 instead of 3 in list

Corrected

Line 85-86, you mention GPR profiles indicated stratified layers in the upper 14m, and the GPR was set to 43 m. How did the layers appear below 14 m? Including a mention from the Thomas et al. 2021 paper here might help with demonstrating feasibility of a longer core record. I would suggest moving the bullet point from the conclusions (line 526) to this location, especially since it is not a conclusion of this manuscript's analysis.

Updated to demonstrate "Near continuous stratified layers were observed in the GPR profiles for the upper 43 m at this site (the maximum time window for the GPR) and bedrock was not detected at this depth. However, the full ice thickness has not been determined."

Line 101, it may be helpful to make this point in the introduction. At present, the ERA5 comparison with the AWS data appears like a separate side-project in the introduction. Making the point that reanalysis products have struggled in Antarctica and that's the motivation for the AWS comparison would justify it a bit better in my opinion.

We will add this to the introduction.

Line 131, were other thinning functions tested in light of not knowing the full depth?

Unfortunately, other thinning functions would require additional in-situ observations (e.g., measured strain rate). This is a limitation of the data that we have, and we have tried to make it clear that this is likely not appropriate for the site. Future deep drilling would require a thorough geophysical survey to determine the most appropriate thinning function.

Is there a reason for not including MSA in figure 2?

The full suite of major ions was measured for this core; however, we decided not to present all the data to avoid overcrowding the in the figure. In addition, the MSA is being utilised by a PhD student in a related project and we decided not to present the data here first. However, we can present all the data in a supplementary figure if this is considered necessary.

Line 165 - greater than 50% of hourly data is a low threshold. How might changing this threshold influence the reported warm/cold bias in ERA5? With such a high correlation between ERA5 and the AWS, I imagine the variability would not be affected too much by altering the threshold, but it could make a difference in the magnitude of temperatures. Might be worthwhile since there is considerable discussion regarding the lapse rate in reconciling these differences.

Good suggestion, we will increase the threshold to explore how the absolute values change.

Line 187 - just to clarify, the lapse rate estimate applied to the AWS greatly reduces the ERA5 warm bias, but the lapse rate estimate applied to the drill site underestimates temperature at the drill site? Is this the reported cold bias mentioned in line 380?

Apologies, there is a typo on line 380. The ERA5 data is warm biased. The temperature difference between the AWS and ERA5 data suggests that ERA slightly over-estimates the temperature.

Line 321 - I am not sure the analysis around SAM and SOI are necessary for this manuscript given how heavily caveated it is due to the short time span. I see the need to demonstrate a longer core's potential in exploring these dynamics, so I can understand its inclusion at this stage. I'm just not sure it adds much at this stage. I would suggest either paring it down by noting much of this analysis takes place in a positive SAM phase and merely mentioning the non-stationarity suggested by Figures 5g-i. Perhaps focus on the Amundsen Sea Low variability at this stage, which could be scaled up to explore SAM, SOI dynamics with a deeper core.

I appreciate that the short timespan makes it hard to draw conclusions about SAM and SOI. Considering this, we will remove the paragraph directly relating to the peter 1st data correlations with SAM and SOI (and remove from r values from table 1).

Figure 5 c and f do not match very well in the details, although in the greater scope the match is sufficient. Does the lack of strong correlation between accumulation and zonal winds extend from post-depositional effects at the drill site or small-scale wind patterns affected topography, coastal proximity, etc.?

Certainly, the post depositional effects will influence the firn core snow accumulation. We will include some text to elaborate on the potential differences as suggested. However, it is a difficult balance this with not over-interpreting such a short record. And as mentioned previously, the pseudo cores may artificially be increasing the correlation coefficients because we are comparing the ERA5 with ERA5. Including both caveats will hopefully explain the potential differences whilst highlighting that we cannot over-interpret the data. The hope is always that a longer record would overcome both these issues and provide more robust correlations.

Line 347: Regarding the sulfate peak at 4.6m, described as the volcanic eruption. Why are there accompanying peaks in Br and Na in Figure 2? I had assumed this was an issue of percolation due to melt. Is the sulfate in Figure 2 just biogenic. If so, how did you tease it out? Would air trajectories help confirm whether volcanic material was transported to your drill site?

This is a good question, and one that is a little difficult to address with the data we currently have. The inclusion of back-trajectories was considered, however, we felt this might be beyond the scope of this study. Our intention is to present the data we have (and be open about its limitations) to demonstrate that this site has potential for future drilling. Whilst leaving room for further work, including back-trajectory analysis with additional proxies. For example, we can look for evidence of tephra in the firn core, however, this would require significantly more analysis and might be best applied to a longer record.

Line 450-454 - the potential to reconstruct blocking variability and the connection with atmospheric rivers are exciting! I think this could be fleshed out a bit more, but that's merely my personal preference.

We agree that this is an exciting aspect of the site location. As mentioned previously, we don't want to overstate this and decided not to expand on the analysis here. While 2006 and 2013 do support the influence of ARs, the 2010 year behaves differently. We feel that more in-depth analysis is required to expand on this connection.

---

## Author Comment (AC2)

Response to Reviewer 2

The reviewer comments in black and our response in blue.

Reviewer 2

Thomas et al. present a new shallow firn record from Peter the 1st island in the Bellingshausen Sea, Antarctica. After developing the timescale of this 15-year-long record, they evaluate relationships among accumulation, melt layers, and d18O with reanalysis products and with a short automatic weather station dataset. The paper lays the groundwork for interpreting a potential deep ice core drilled at the site, which is expected to span about 200 years. With some organizational restructuring and some clarifications, the paper is worthy of publication in a journal such as Climate of the Past.

Thank you for your time reviewing this manuscript, your comments and suggestions are appreciated.

Major comments:

The development of the age model is foundational to the paper, as the accumulation rate derives from this and is used in the subsequent correlations. I am surprised by the tiny size of the seasonal variations in the major elements, and am curious what nssCa would show. Ice core records from West Antarctica typically show well-resolved annual dust peaks, so it seems like Ca could be really helpful here. If the data were collected, why are they not included? At any rate, most of the annual picks seem reasonable visually; however, the pick between 2005 and 2006 (Fig. 2) aligns with troughs in most of the major ions, rather than peaks as for most years – and there are peaks to either side of this pick that are not used. Why is this?

Annual layer counting is a powerful tool for ice core research; however, it has its limitations. While many proxies display clear seasonality at inland locations, the seasonal deposition at lower elevation, coastal and island sites can be quite different. This is likely a result of the lower elevation of the arriving air-masses, which contain a greater contribution of local (marine) species and compounds, but less contribution from long-range sources (e.g., dust). Sulphate is a good example of this, and has been explored recently in Emanuelsson et al., 2022.

In the case of Ca, the record does display a seasonality. However, it looks almost identical to Na. The two records are highly corelated (r=0.8, n=249), suggesting that a significant contribution of Ca is from marine (not continental) sources. When comparing with the seawater ratio (based on Cl), nssCa accounts for just ~28% of the Ca.

Second point related to the timescale is that the potential tie-points (Puyehue Cordon Caulle and the two melt years, 2006 and 2013) need to be tied in earlier in the text and not saved for the discussion. This text should be part of section 3.1 so that the reader can fully evaluate timescale development with all information in the same place. The apparent PCC peak should be highlighted in the figure. The 3-point smoothing draws the eye away from this prominent peak, but I think it would be helpful to make it more obvious. I would suggest instead of a dashed line, using a thin solid line for the full-resolution data. In the Fig. 2 caption, clarify that summer peaks are defined as Jan. 1 (or alternate date, as chosen by authors). Labeling the years between the lines (rather than directly above each line) is a bit un-intuitive for me, so it would be helpful to define exactly what the black lines represent.

Thank you for all the great suggestions. We have made the updates to figure 2 as suggested.

We have also moved the text from the discussion (4.1) to the main results section (3.1).

I also agree with Reviewer 1's main points, but as these concerns have already been raised, I will address other issues.

Minor comments:

Line 103: Please include actual time spans of data collection. As written, it implies one year – but the actual record is much shorter.

Updated

Line 107: Needs a transition here.

Updated.

Line 120: This seems like a small subset of the major ion suite that is typically measured. Why were Mg2+, Ca2+, NO3-, NH4+, Cl-, etc. excluded from the analysis? Or were they analyzed but not included here? It seems like nssCa could help support the timescale development, so it would be nice to see it included. If the other data are to be presented elsewhere, then include a statement to that effect. Okay, re-reading this I see in section 3.7 that other major ions were also measured. Please update the methods to include all measured ions, and present the data in Fig. 2 to reinforce timescale picks, as suggested above.

The full suite of data was analysed, however, we selected only the optimal data for figure 2. This was to avoid over-crowding and repetition. E.g., most of the cations (Na, Mg, Ca) behaved the same and adding individual curves didn't add much. However, we can either update figure 2 with the additional data or provide a supplementary figure containing all the major ions.

Regarding nssCa, this might not be appropriate for an island location. We can assume that the continental fraction would be extremely low.

Line 129: correct to "site's"

Corrected

Line 130: "ensuring" seems like an odd word here. Wouldn't "assuming" be more appropriate? And why is the assumption that the firn core only samples the upper 10% of the ice cap's thickness? I think a little more explanation of the reasoning here would be helpful.

This was to justify the use of the Nye model, which is appropriate for the upper 10% of the ice sheet. I have rephrased the sentence to better reflect this.

Line 131: Include some uncertainty estimates based on other thinning models. How much could that 130 m estimate reasonably change based on the model you use?

This is a little difficult to explore with the limited data that we have. While we accept that the Nye model is probably not appropriate, the lack of in-situ observations (e.g., strain rate) limit the other thinning functions that could be applied. We have attempted to estimate the error range by altering the annual snow accumulation variable. To avoid over-interpreting the short record that we have, we would rather expand on the thinning function using in-situ geophysical information in the future.

Line 161: Should be "provide"

Corrected.

Line 172: the circle appears purple, not blue. For this figure, the black/gray/dark purple coloring makes it a little tricky to visually resolve the lines. I would suggest using more of a royal blue (i.e, more contrast with the black and gray) for the ERA5 data.

The figures have all been updated to RGB colour mode, to provide better contrast (including brighter blues).

Line 191: There is a fragment here that needs revision.

Updated.

Line 179 and the paragraphs that follow: Here it is stated that "a lapse rate is applied." However, it is unclear what rate is actually used and why. This whole section (3 paragraphs about lapse rates) could use reorganization and clarification.

Paragraphs updated and clarified.

Line 200: This section 3.3 seems like a subheader of the section above, as the two topics are closely related.

These sections can be combined to include 3.3. as a sub-sections of 3.2.

Line 207: Replace "meltiest" with "heaviest melt" or similar

Noted, although it is a good word.

Line 211: There does not appear to be a melt layer in Dec 2006 (Fig2). Can the authors clarify or explain why this may be?

Given the uncertainties of the dating, it is hard to accurately assign melt to a single month. However, there is a melt layer in the summer of 2006/2007. In figure 2, this appears coincident with the year marker (black line), which we assume spans the warmest months of the year (e.g., December 2006/ January 2007).

Line 231: Another fragment here that needs to live its full life as a sentence.

Corrected

Line 239: This paragraph discusses accumulation, which is listed as the header of the next section. This whole part of the discussion needs reorganization in its structure and labeling. Part of this could be to remove the heading of 3.5: Accumulation, and allow the discussion of accumulation to follow smoothly from heading 3.2 as these paragraphs are so closely related.

Agreed, these subsections can be combined.

Line 302: Change "warm-moist" to "warm, moist"

Changed

Fig. 5: Please label the y-axes next to panels d and g "ERA 5 P-E (year range)" so it's clear exactly what the plots show.

Updated

Line 329: Change "temporality" to "temporally"

Updated

Line 350: Please cite Koffman et al 2017 for the PCC reference horizon

Absolutely, sorry for the omission.

Section 4.1 can be moved into the first discussion of the age scale.

Agreed

Line 365: Would "ice cap" be more appropriate than "ice sheet" here and throughout the paper?

Line 384: looks like a typo in the citation; colon should be a semicolon

Line 403: Correct to "or has been lost"

Line 422: "island's"

Line 440: awkward syntax in this sentence, please revise

Line 448: Please clarify which continent (South America or Antarctica" and change "to" to "the"

Line 466: Change "is" to "can be"

Line 478: correct to "is a subject for further study"

All points above updated and corrected.

Paragraph lines 477-484: This paragraph needs to be reorganized. Give the information and observations about melt years and then end with saying that the topic will be studied further – but leading with this does not fit well with the additional details given immediately afterward.

Agreed.

References:

Added

Koffman, B. G., Dowd, E. G., Osterberg, E. C., Ferris, D. G., Hartman, L. H., Wheatley, S. D., Kurbatov, A. V., Wong, G. J., Markle, B. R., Dunbar, N. W., Kreutz, K. J., & Yates, M. (2017). Rapid transport of ash and sulfate from the 2011 Puyehue-Cordón Caulle (Chile) eruption to West Antarctica. J. Geophys. Res. Atmos., 122. https://doi.org/10.1002/2017JD026893

---

## Author Response (AR1)

Response to editors

Dear editors

I have attached a revised manuscript, which includes all the recommended changes as identified in the previous response to reviewers. I thank the two reviewers again, who provided constructive feedback and recommendations for minor corrections.

In particular, the point regarding my chosen threshold for comparing with the ERA5 and AWS data. In the original manuscript I used a threshold of 50%. However, following the reviewer suggestions I compared the offset between the ERA5 and AWS when just using data where over 90% of the hourly data was available. The correlations between the two datasets remained the same, but the offset between the two records increased from 0.93 to 1.07 °C.  This offset has been updated throughout the text.

Responding to the specific point (1) by the editor, I can confirm that I have updated the dating figure (Fig. 2) to include a larger suite of ions (including MSA, Ca and Cl). The intension is to demonstrate the coherence between all the species used to identify the seasonal peaks.

Regarding the second point (2) I find this is not possible. I would argue that the ability to "unequivocally" identify the source of volcanic deposition and timing of melt layers is not possible in any ice core. As stated in my review, the identification of tephra is not a simple task, and is certainly beyond the scope of this study.

The aim of this paper is to explore the potential that chemistry and isotopes from Peter 1st island are related to climate variability. I fully acknowledge the limitations of using a short proxy record in the text. However, given the unique location I feel there is value in presenting the record, despite some unavoidable uncertainties.

I hope the editor will agree that I have made sufficient amendments to the manuscript.  I hope the editor can now agree with the two reviewer's recommendations as minor revision and accept it for publication in this special issue of climate of the past.

Kind regards

Liz

---

## Author Response (AR2)

We thank the editor and the reviewers for their time evaluating this paper. Based on their recommendations, we have made significant changes to the manuscript.

Firstly, we have undertaken additional analysis to support the age-scale. We have identified crypto-tephra-like shards believed to originate from the Puyehue-Cordón Caulle eruption in 2011. In addition, we have expanded the section on melt, to highlight the good agreement between annual melt and positive degree days which further supports the age-scale. Finally, we have removed the pseudo core data, to avoid the circular argument of comparing reanalysis data.

We hope that this additional information, and more conservative interpretation approach, will alleviate the reviewers concerns and make this acceptable for publication in Climate of the Past.

Detailed response below, with reviewer comments in italics.

*Reviewer 1:*

- *My main concern remains with the presentation of the dating section. Given the short record, it requires a more detailed explanation and justification. Specifically, there is a lack of direct comparison of this record to other firn/ice/snow samples (cores) obtained in the Antarctic Peninsula.*

We were cautious about making comparison with such short records that are more than 600km from the site. However, for the benefit of the reviewers below is a comparison between Peter 1st and the closest continental site, Rendezvous. While the overlap between the two records is just 10-years, and arguably too short to determine meaningful correlations, the seasonal deposition of $SO_4^{2-}$ does provide some supporting evidence for our age-scale.

- *According to Koffman et al., 2017(https://doi.org/10.1002/2017JD026893) deposition and transport of the ash from Puyehue-Cordón Caulle in 2011 happened during 2-3 weeks in June. On Fig.2 this peak is partly located in 2012. Could that be just a seasonal SO4? How this peak is different to the peak in e.g. 2005 (Fig.2). Please clarify.*

[Figure]

Figure R1. Comparison of $SO_4^{2-}$ (blue), nss $SO_4^{2-}$ (black) and deuterium excess data (red) in Peter 1st (top) and Rendezvous (bottom), which are located ~600 km apart.

Firstly, there is no evidence of a significant $SO_4^{2-}$ peak in the winter layer immediately following the Puyehue-Cordón Caulle eruption in Rendezvous. In contrast to snow pit observations in West Antarctica (Koffman et al., 2017). Instead, we observe a broader $SO_4^{2-}$ peak during the spring and summer 2012 layer at Rendezvous. The Rendezvous core was drilled in January 2013, so we have confidence that the June 2011 eruption does not generate a significant winter peak at this site. We conclude that the lower sampling resolution (and potential influence of melt at Peter 1st), likely account for the differences between the West Antarctic snow pit and the ice core records. Therefore, we maintain that the peak in $SO_4^{2-}$ occurs during summer 2012, likely from a combination of volcanic $SO_4^{2-}$ (deposited during early spring 2011) overlain on the summer biogenic $SO_4^{2-}$ deposition.

In addition, as observed at Peter 1st, the Rendezvous record contains elevated $SO_4^{2-}$, between years 2004 and 2006. The increased variability at both sites further supports our age-scale and suggests a similar (yet unknown) source. As observed previously, the pattern of variability, magnitude, and exact timing of $SO_4^{2-}$ peaks in Antarctic ice cores may vary (e.g. [Emanuelsson et al., 2021]). This is especially true when trying to compare sites which are separated by over 600 km of open ocean and contain relatively high background biogenic $SO_4^{2-}$.

- *Additionally, more clarification is needed on the criteria used for annual layer placement, particularly for years such as 2012/2013!, 2009/2010, or 2007/2008. In Figure 2, some lines do not align with any peaks. Without additional evidence, the annual layer counting could yield results ranging from 7 to 16 years. It is crucial to provide independent evidence to support the assumed correct dating. Alternatively, you can try to provide different possible dating of the core and then check the results and correlations (in supplementary).*

We have now included the deuterium excess in figure 2, to further demonstrate our selection of individual years. We hope that the revised figure 2 helps to explain our depth horizon choice.

We have also undertaken a more in-depth evaluation of potential volcanic eruptions, to further constrain the age-scale (see below). We now have evidence for crypto tephra in the layer corresponding to 2011/2012. This independently supports our age-scale, providing a reference horizon for the Puyehue-Cordón Caulle eruption in 2011.

- *Can you please present nssSO4 and nssCa data to compare with other records?*

Given the maritime location, the contribution of marine $SO_4^{2-}$ is high. Traditional equations for nssSO$_4$ (e.g. [nss $SO_4^{2-}$] =[ $SO_4^{2-}$] -(0.25*[Na$^+$])) result in a largely negative record. The nss $SO_4^{2-}$ peak is shown in Fig R1 (black), for the reviewers. However, we remain unconvinced that extracting the nss $SO_4^{2-}$ component from such a low elevation maritime location is very informative.

- *While the authors correlate the record with meteorological data, these data are not presented. It would be helpful to include, for example, PDD temperatures over the entire period to demonstrate notable years such as 2013 and 2006.*

We have now included a more detailed comparison of the meteorological data. We present the annual melt thickness alongside the number of positive degree days (PDD) and daily maximum temperature. This highlights the close agreement between melt and temperature (r=0.73), which further supports our age-scale. The data is presented in a revised, which now also includes the line scanned image of the melt feature previously presented as a supplementary figure.

- *Regarding sulfate peaks, authors suggest volcanic eruption as a potential source, but additional evidence is needed. Given the low concentrations and the presence of high calcium levels (suggesting that this layer is alkaline and not acidic), it is important to consider other independent data such as cryptotephra.*

To independently validate our assigned volcanic reference horizons, we have now scanned the particulate material from this core. The presence of several cryptotephra particles in the layer corresponding to 4.6-4.9 m confirm the volcanic origin of the sulphate peak at this depth, and further support our assignment that this is the 2011 Puyehue-Cordón Caulle eruption.

We have referenced the work of Koffman et al., 2017 as supporting evidence that this material was likely deposited after June 2011.

- *Regarding snow accumulation, the reasoning for accounting for annual layer thinning for the 14 m firn core should be explained. If such corrections are necessary, references, formulas, and the calculated effect should be provided. It is worth noting that snow and firn compaction does not alter the water-equivalent of layer thickness since there is no lateral extension (Nye, 1963 DOI:https://doi.org/10.3189/S0022143000028367). Please clarify.*

We have used the Nye model to account for thinning at this site, failing to cite the appropriate paper was an oversight. It has now been included.

- *L.296. Regarding the removal of the two most melt-affected years (2013 and 2006 CE), further clarification is needed on the correlation with winter average 2m temperature. It's unclear what "respectively" refers to, and the correlation value of r=0.66 is not presented in Table 1.*
- *Additionally, the rationale for correlating annual isotopic composition with winter temperature should be justified, considering precipitation distribution. By removing two years, you make the record even shorter and it is not clear that 2006 isotopic record was affected that much by melting so it should be removed.*

Firstly, we have decided to only include correlations with the annual average data. This has now been revised in figure 6. We also no longer remove the most melt effected years from our comparisons or corelations. While we think the melt has impacted the record, we agree that removing them makes the record too short to draw meaningful significance. Accordingly, we have removed table 1 which is no longer required.

Reviewer 2:

*Minor:*

*Lines 25-27 and 65-72, Introduction and Discussion: This sentence would fit better within the abstract if the clauses were flipped.*

*Lines 65 to 72 demonstrate that evaluating the skill of ERA5 on Peter 1st Island and comparing the firn core proxies with ERA5 data are two goals of the paper.*

Thank you for the suggestion, we have updated the abstract text with the revised sentence to better reflect the goals of this study.

*Line 32 and section 2.1: How do these crevasses influence the ice cap as a possible drilling location? Is there a location that is above the crevasses and may not be interrupted by these weak points that can influence the ice layers? If one of the main goals is to establish the suitability of the site for future deep ice core drilling, how often does this heavy cloud cover occur? Obviously, drilling in such a remote location with difficult logistics can influence the site location. Is the site of the short core a possibility for drilling the longer core?*

In section 2.1, we present a geographical setting for the firn core presented in this study. Based on the data we collected from the firn core, and the short GPR transects, we suggest there is potential for future deep drilling. However, any future deep drilling would be subject to a far more extensive geophysical survey to identify a site free from crevassing.

*Lines 31 and 32: Include the coordinates of the island or refer to the map with the location (Figure 1).*

Added.

*Line 40 and throughout the paper: The use of the acronym "AP" for the Antarctic Peninsula hinders rather than helps the reader. Please use the full name throughout the paper, rather than relying on the acronym.*

All changed.

*Line 44: How long did this pause in warming occur? Are the temperatures at a record level, or is the rate of warming at records levels (or both options)?*

The temperatures are at record levels. This has been clarified in the text.

*Line 69: I think you wish to complete this sentence with "provide a unique opportunity to …." Rather than "unique op."*

Corrected.

*Lines 94-125: Did the samples melt at any time during the transit? How were the samples distributed between the teams? Were all samples brought back to the British Antarctic Survey?*

The samples remained frozen. The ice cores were transported in a -25°C shipping container to the BAS laboratories. All samples were sub-sampled at BAS.

*Line 330: As you are using brackets, the word "concentration" is not necessary. If you choose to keep this word, then change it to the plural "concentrations".*

Removed.

*Section 3.7: Substantially more information on how the "pseudo core" was derived is necessary and/or omitting the entire section that uses the pseudo core data. While the authors are clear that the pseudo core can create circular reasoning, the pseudo core does not grant sufficient gains in understanding that outweigh this circular reasoning.*

Based on this, and other reviewer comments, we have decided to remove the section relating to pseudo cores. We feel that the potential circular reasoning is detracting from the aim of the study. Discussion section 4.4 has been shortened to just provide an overview of expect drivers of variability based on current literature.

*Technical corrections:*

*Line 28: Why say "invaluable" when you can simply state "valuable"?*

*Line 32: Replace "islands'" with "island's".*

*Line 57: Change "last Glacial Maximum" to "Last Glacial Maximum".*

*Line 228: Change "when temperature have exceeded" to "when temperatures have exceeded".*

*Line 230: Change "To test this" to "To test this discrepancy".*

*Line 260: Change "However, this suggest" to "However, this suggests".*

*Line 315: Change "reference" to "references".*

*Lines 462, 535, 555 and 636: Change "Peter 1st island" to "Peter 1st Island".*

*Line 569: Change "This is corroborated" to "This warming is corroborated".*

*Line 578: Change "Mean annual temperature of" to either "Mean annual temperatures of" or "A mean annual temperature of".*

All technical corrections have been made.

---

## Author Response (AR3)

Response to reviewer:

Thank you for reviewing this manuscript again. The proposed suggestions have been made to the test, with the answers to questions included below.

Line 94: Is the -25 C shipping container a stand-alone container that is somehow always kept at the specific temperature? Or is the container once the core is transported from the island to the subsequent transportation?

This is a standard dual-compressor reefer shipping container. The temperature is set and then monitored throughout the voyage.

This section has been expanded to provide more detail about the shipping, and to confirm that the temperature was maintained during transport.

Lines 101-102: Were the cores cut into the discrete samples in the field or in the lab?
In the lab. The following added for clarification:

"Subsequent sub-sampling of the cores was undertaken in the -25°C cold laboratories."

Section 2.6: While I understand wanting to place this section within the methods, it makes more sense to first describe the age scale and then describe the uncertainty. The current format describes the uncertainty before the age scale, which makes the reader then scramble to try to find the age scale. (Granted, the age scale is in the subsequent section, but it is still necessary to describe the age scale first).

As suggested, this section has now been removed. The text regarding uncertainty bars is only relevant for figure 6 and has been included in the caption.

The section about age-uncertainty has been moved to section 4 and the statistical significance sentence moved to section 2.2.

Line 167: Do you perhaps mean November to December (2 months of a summer peak) rather than December to November? Or is the dash really a minus sign and therefor you are examining the difference between the December concentrations and the November concentrations? (Which is highly unlikely as you are looking at seasons rather than specific months).

Yes, I meant November-December (referring to the two summer months). This has been updated.

Lines 260-277, lines 380-385: Line 263 suggests the possibility of a cold bias throughout ERA5, while lines 276-277 suggests the possibility of a warm bias in ERA5 "between 0.52 to 0.67 °C at this location". Are you suggesting that both of these biases occur simultaneously? For example, a cold bias for all temperatures except for between 0.52 and 0.67 C on the island? If so, this reasoning needs to be expanded. In lines 260-277 I was proposing two possible scenarios to explain the discrepancy between ERA5 temperature and visible melt. Lines 380-385 suggest that the warm bias is an increase of 0.52 to 0.67 °C rather than between these two temperatures. Please clarify.

The text has been updated to clarify that the data indicates that ERA5 is warm biased at this site.

Line 343: Change "sites" to "site's"

Corrected.

Lines 362-363: Although Thomas et al., 2021 is cited here, more explanation needs to be included in this paragraph regarding the "previous estimate". How was the previous estimate calculated? If the previous estimate is from the P-E from ERA5 then evaluating ERA5 becomes circular reasoning throughout the entire paper.

This was based on the Herron and Langway densification model using the measure density profile. This section has now been expanded to include a more detailed description.

Lines 450-455 are speculative and can be reduced to one to two sentences to minimize the speculation.

Sentences combined to reduce speculation.